# Penicillin-Binding Proteins in *Streptococcus agalactiae* and Their Association with Reduced Penicillin Susceptibility: An Overview

**DOI:** 10.3390/antibiotics15010031

**Published:** 2026-01-01

**Authors:** Leonardo Nagao Ferreira, Bruna Alves Pimentel Hecht, Louisy Sanches dos Santos, Prescilla Emy Nagao

**Affiliations:** 1Souza Marques Technical Education Foundation, Souza Marques School of Medicine, Rio de Janeiro 22793-140, RJ, Brazil; leo.nagao.ferreira@gmail.com; 2Laboratory of Molecular Biology and Physiology of Streptococci, Institute of Biology Roberto Alcantara Gomes, Rio de Janeiro State University (UERJ), Rio de Janeiro 20550-013, RJ, Brazil; bruna.hecht@hupe.uerj.br; 3Laboratory of Diphtheria and Corynebacteria of Clinical Relevance, Faculty of Medical Sciences, Rio de Janeiro State University (UERJ), Rio de Janeiro 20551-030, RJ, Brazil; santos.louisy@uerj.br

**Keywords:** *Streptococcus agalactiae*, penicillin-binding protein, reduced penicillin susceptibility

## Abstract

*Streptococcus agalactiae* are bacteria that can cause a range of infections, some of them life-threatening. Currently, antimicrobial resistance has become a global problem that puts public health at risk. Despite the widespread use of β-lactams, penicillin remains the first-line antimicrobial for the treatment of invasive *S. agalactiae* infections. However, reduced susceptibility and resistance to penicillin have been identified in several countries. Penicillin-binding proteins, mainly PBP2X, have been associated with reduced susceptibility to β-lactams in streptococci. The aim of this review is to summarize currently published data on penicillin-binding proteins in *S. agalactiae* and penicillin susceptibility, highlighting the increasing number of strains with reduced susceptibility and resistance to penicillin commonly used in the prophylaxis and treatment of invasive infections by this pathogen. Data on invasive *S. agalactiae* strains with high levels of penicillin resistance have been found in Japan, the United States, Canada, and Africa. The data on antibiotic resistance are alarming and require increased monitoring of strains with reduced penicillin susceptibility, as well as preventive control measures to avoid the spread of resistant mutant strains.

## 1. Introduction

*Streptococcus agalactiae* (group B Streptococcus, GBS) is widely recognized as a leading cause of neonatal sepsis and invasive disease in the elderly or people with comorbidities [1]. Penicillin remains the first-line antibiotic for the treatment of *S. agalactiae* infections and is crucial for preventing vertical transmission to the newborn through intrapartum antibiotic prophylaxis (IAP) [2]. However, isolation of *S. agalactiae* presenting reduced penicillin susceptibility (RPS) has been reported since 2008 [3] and has become a matter of concern worldwide.

The emergence and progression of antimicrobial resistance constitute one of the most significant dangers to public health, as highlighted in clinical investigations supervised by the World Health Organization (WHO). RPS *S. agalactiae* strains are defined based on antimicrobial susceptibility testing. Both the European Committee on Antimicrobial Susceptibility Testing (EUCAST) and the Clinical and Laboratory Standards Institute (CLSI) provide standardized guidelines for performing antimicrobial susceptibility testing and for interpreting clinical breakpoints for *S. agalactiae*. EUCAST has established an epidemiological cut-off value (ECOFF) for penicillin in *S. agalactiae* of 0.125 mg/L, as well as zone diameter breakpoints of >18 mm [4]. According to CLSI, isolates of *S. agalactiae* with RPS are defined as those with a minimum inhibitory concentration (MIC) > 0.12 µg/mL [5]. Isolates with MICs above this threshold are considered to have acquired resistance mechanisms.

In *S. agalactiae*, the RPS has been attributed to the acquisition of mutations in genes encoding the penicillin-binding proteins (PBPs) [6,7], enzymes that catalyze the biosynthesis of bacterial cell wall peptidoglycan. These mutations result in the production of PBPs with reduced affinity for β-lactam binding. However, significant gaps remain in our understanding of the functional impact of individual or combined PBP mutations, therapeutic failure resulting from small increases in MIC, and the evolutionary dynamics that may drive the worldwide spread of RPS *S. agalactiae* strains.

Therefore, this review aimed to gather available epidemiological data and current knowledge on the most relevant PBP mutations associated with RPS and penicillin resistance (PR) in *S. agalactiae* strains, contributing to identifying priorities for future research and improving the clinical management of *S. agalactiae* infections.

## 2. Penicillin-Binding Proteins (PBPs)

PBPs comprise two classes based on their molecular weight and enzymatic activity (glycosyltransferase and/or transpeptidase). Low molecular weight PBPs (LMM class C) are monofunctional enzymes, such as carboxypeptidases involved in peptidoglycan remodeling, while high molecular weight PBPs (HMM class A and B) are responsible for peptidoglycan polymerization and its insertion into the pre-existing cell wall [8]. The N-terminal domain of HMM PBPs class A is responsible for glycosyltransferase activity, catalyzing the elongation of non-crosslinked glycan chains. After lipid II (disaccharide-pentapeptide attached to the pyrophosphate-tethered undecaprenyl tail) is inverted to the periplasmic side, glycosyltransferases polymerize the sugar chains (Figure 1A), while the penicillin-binding C-terminal domain has transpeptidase activity, catalyzing the cross-linking of peptides between two adjacent glycan chains. PBP recognizes the terminal portion of D-alanine, catalyzing the attack of the carbonyl group of the penultimate D-alanine by the lateral amino group at position (3) of an adjacent chain (4–3 cross-link; Figure 1B). In class B, the N-terminal domain plays a role in cell morphogenesis through interaction with other proteins involved in the cell cycle [9].

PBPs possess a highly stable covalent complex through the serine in their active site, where penicillin and other β-lactams bind irreversibly, forming a stable acyl-enzyme intermediate that permanently inactivates the enzyme [10,11]. The catalytic serine attacks the carbonyl group of the β-lactam ring, promoting ring opening and the formation of a stable covalent acyl-enzyme complex that impairs transpeptidation, leading to peptidoglycan cross-linking failure and bacterial death [12]. Altered PBPs, resulting from multiple homologous recombination events between genes of closely related species (mosaic genes), combined with additional point mutations, have been described in pneumococci and oral streptococci, where amino acid substitutions represent more than 10% of the primary sequence [7]. Identifying the amino acid alterations involved in resistance, combined with structural information, will provide a better understanding of the enzymatic function of PBPs and the development of new inhibitors. However, mosaic sequences of *pbp* genes represent a challenge for classification and organization. Comparison of nucleotide sequences from susceptible strains reveals that these sequences exhibit levels of polymorphism similar to other loci, with one or two amino acid substitutions along the protein [6]. Mosaic *pbp* genes also exhibit sequence blocks that approximate the non-mosaic alleles in PBP2B, PBP1A, and PBP2X [6,13,14]. These blocks can encompass various regions of the transpeptidase domain or even a large part of the extracellular domain, the magnitude of which suggests that these divergent sequence blocks originate from other streptococcal species to escape protein selective pressure [6,15,16]. *S. agalactiae* produces three bifunctional class A PBPs (PBP1A, PBP1B and PBP2A) and two monofunctional class B PBPs (PBP2X and PBP2B) [17]. However, few studies report the biological functions and differences between PBPs in *S. agalactiae*. Therefore, to fill this gap, the general differences and biological functions of the PBPs analyzed in different microorganisms will be described.

The origin of sequence blocks in *pbp* mosaic genes is still unknown, with some exceptions for PBP2X. A study revealed that a family of mosaic *pbp2x genes* occurred in *Streptococcus pneumoniae*, *Streptococcus mitis* and *Streptococcus oralis,* demonstrating independent inter- and intraspecies recombination events [18]. These data gave rise to the hypothesis that commensal streptococci, such as *S. mitis* and *S. oralis,* acquired resistance through point mutations by exposure to β-lactam treatment for diseases, being subsequently exchanged between related streptococcal species. PBP2X depletion in *S. pneumoniae* promoted elongated and swollen cells with pointed ends [19]. Resolution 3D structured illumination microscopy (3D-SIM) showed the exceptional localization of PBP2X in the centers of septa in cells at intermediate to late stages of division, suggesting that the proteoglycan transpeptidase cross-linking activity of PBP2X has a biological function during septum formation and proteoglycan remodeling [20]. Phylogenetic analysis showed that *S. pyogenes* strains of the same *emm* type and the same amino acid substitution in PBP2X were clonally related by descent, indicating that strains with alterations in PBP2X can spread to new hosts and cause invasive infections [21].

PBP1A, also called PonA, and PBP1B proteins contain a non-catalytic domain involved in the regulation of glycosyltransferase and transpeptidase activities and are essential for cell viability in *S. agalactiae* and *S. pneumoniae* [10,17,22]. PBP1A is preferentially involved in proteoglycan synthesis during cell elongation and plays an important role in the resistance of *S. agalactiae* to antimicrobial peptides, potentially representing a widespread mechanism for evading the human innate immune response [23]. PBP1A was important for lung proliferation and resistance to killing by polymorphonuclear leukocytes and alveolar macrophages in a neonatal mouse model of *S. agalactiae* infection [24]. The mutant *S. agalactiae* strain *ponA* also showed reduced virulence in a murine sepsis model compared to the wild-type strain [25]. PBP1B is involved in septation, being found in high concentrations in the mid-cell region during cell constriction. FtsN, a divisome-specific protein, binds to PBP1B, stimulating its glycosyltransferase biosynthetic activity, contributing to the synthesis and overall regulation of septal peptidoglycan during cell division [26].

PBP2 is involved in cell division, particularly at its location in the septum. Certain bacteria, such as methicillin-resistant *Staphylococcus aureus* (MRSA), where the first crystal structure of a soluble form of PBP2A was published in 2002, acquire PBP2A [27]. Crystallographic analyses showed that PBP2A, the *mecA* gene product, could maintain transpeptidase activity and resist β-lactam binding, maintaining a closed conformation of the active site. Thus, interaction with a second molecule (e.g., muramic acid, peptidoglycan, or ceftaroline) at an allosteric site induced a conformational change, allowing the active site to open for catalysis [28]. The ability of ceftaroline, a β-lactam antibiotic, to stimulate the allosteric opening of the active site, leading to the inactivation of PBP2A by a second β-lactam, will allow future studies to obtain allosteric inhibitors based on the structure of PBP2A.

PBP2B is a class B specific to cell division with transpeptidase activity [29]. In *S. agalactiae*, researchers have identified that the polymerization of the glycan chain by the glycosyltransferase activity of RodA, a well-characterized SEDS (shape, elongate, and divide) family protein, is activated by PBP2B, generating the production of longer peptidoglycan fragments and faster depletion of the substrate lipid II [30].

Interestingly, Nishimoto and co-authors suggested that resistance to β-lactams in pathogenic streptococci is restricted to naturally competent species through intra- and interspecific recombination due to in vivo fitness trade-offs from de novo mutations in PBPs. The authors demonstrated that serially transformed recombinant strains efficiently integrate into the DNA of resistant oral streptococci, acquire resistance and tolerance to β-lactams, and maintain virulence in an in vivo model, indicating that de novo mutations are of primary relevance [31]. However, other components of the peptidoglycan synthesis machinery likely interact with PBPs directly or indirectly and may be modified in response to changes in the PBPs. All these facts contribute to the difficulties in experimentally verifying the impact of individual mutations within PBPs on penicillin resistance. The availability of larger amounts of genomic data for RPS and commensal streptococci will facilitate obtaining more information about the complex genetic network between species and will unravel mutations in components involved in the expression of penicillin resistance. These observations raise questions about the different roles that PBPs may play in cell wall formation and the possibility that their functions may vary slightly depending on the nature of the bacterium, the local environment of the cell, and the stage of the cell cycle.

## 3. Mutations in Penicillin-Binding Proteins (PBPs)

Identifying amino acid substitutions relevant to the reduction in affinity for specific PBPs is a difficult task because, due to the recombination process, neutral substitutions were probably imported along with those that confer antibiotic resistance. However, the number of likely important substitutions was proposed based on their absence in susceptible strains, presence in many resistant strains, and proximity to catalytic motifs [32]. RPS or resistance to β-lactam antibiotics may be due to reduced access to PBPs, low binding affinity to PBPs, or β-lactamases that promote hydrolysis of the β-lactam ring [12,33]. PBPs contain three essential conserved sequences (SXXK, SXN, and KT(S)G) involved in the transpeptidation mechanism, crucial for bacterial cell wall synthesis. Alterations in these motifs or adjacent areas can decrease the binding affinity of PBPs to β-lactam antibiotics [34]. This resistance mechanism often arises through homologous recombination between related bacterial species in environments where antibiotic pressure is significant. In *S. agalactiae*, mutations in genes encoding PBPs have been associated with increased antibiotic resistance. Comparative phylogenetic studies have indicated that multiple RPS *S. agalactiae* genetic lineages have developed independently due to the accumulation of mutations, particularly in the *pbp2x* gene, which plays a significant role in reducing the organism’s susceptibility to β-lactams [3,34,35,36]. This understanding highlights the genetic adaptations in PBPs that contribute to antibiotic resistance, complicating the treatment of infections caused by *S. agalactiae.*

Several mutations have been identified in *S. agalactiae*, resulting in amino acid substitutions close to the conserved active site (Table 1), including V405A and/or Q557E of the PBP2X transpeptidase [35,37]. Four distinct classes (I–IV) have been proposed based on specific PBPs exhibiting different amino acid mutations. Important amino acid substitutions in the active site of PBP2x have been established by kinetic and structural studies. Mutation in two residues in the serine region downstream of the active site altered the orientation of the serine hydroxyl group, decreasing the affinity with β-lactams [38]. Furthermore, high-resolution analysis of PBP2X and multiple sequence alignments demonstrated an amino acid cluster lining a small water-containing cavity, presumably involved in PBP2X resistance to β-lactam antibiotics. Site-directed mutation in these amino acids significantly reduced the affinity for β-lactams [39].

Mutations and substitutions close to the active sites of the PBP2B, PBP1A, and PBP2A enzymes are also considered critical [40]. Deletion of *pbp1a* or *pbp1b* genes resulted in increased sensitivity to penicillin and ampicillin, suggesting that PBP1A and PBP1B play an important role in transpeptidation relative to PBP1A or PBP2B when *S. agalactiae* strains are under stress due to β-lactam antibiotics. Deletions in the *pbp1a* and *pbp2b* genes also promoted the growth of bacterial cells in significantly longer chains, suggesting that the mutant strains exhibited a defect in cell separation during the stationary phase [17]. The emergence of *S. agalactiae* isolates with reduced sensitivity or resistance to penicillin is concerning.

**Table 1 antibiotics-15-00031-t001:** Reduced susceptibility to penicillin amongst *Streptococcus agalactiae* isolates and the amino acid substitutions in *pbp* genes.

Author, Year [Reference]	Penicillin MIC	Mutations Identified				Country (Period)
		*pbp2x*	*pbp2a*	*pbp2b*	*pbp1a*	
Chu, 2007 [41]	0.25 (*n* = 1)	ND	ND	ND	ND	Hong Kong (2005)
Kimura, 2008 [3]	0.25 (*n* = 8) 0.5 (*n* = 5) 1.0 (*n* = 1)	**Q557E, V405A**	ND	ND	ND	Japan (1995–2005)
Nagano, 2008 [35]	0.25 (*n* = 1) 0.5 (*n* = 6) 1.0 (*n* = 2)	**S353F, A374V, F395L, V405A, A400V, R433H,** **H438Y, A514V, Q557E, G648A, T77I**	E63K, **T175I,** **L285F, Y236C**	**V80A, Y262N,** **G539E, T567I, G613R**	**L45P, N163K,** **Y470F, G527V, N723S**	Japan (2003–2004)
Gaudreau, 2010 [42]	0.25 (*n* = 1)	I377V, G627V, **N575D**	S453N, N682D	V625I, **P278L**	**T526A**	Canada (2004–2007)
Longtin, 2011 [43]	0.5 (*n* = 1)	**G371D**	**E636G, S644F, S676F**	ND	**T546P**	Canada (2008)
Nagano, 2012 [44]	0.25 (*n* = 10)	**V405A, F395L, R433H, H438Y, G648A**, I377V, V510I	NT	**T567I**	ND	Japan (2007)
Nagano, 2014 [45]	0.5 (*n* = 2)	A400V, **V405A**	ND	Q557E, **T567I**	ND	Japan (2011–2012)
Seki, 2015 [46]	0.25 (*n* = 19) 0.5 (*n* = 20) 1.0 (*n* = 6)	**V405A, Q557E**	NT	NT	NT	Japan (2012–2013)
Morozumi, 2016 [47]	0.125 (*n* = 5) 0.25 (*n* = 3) 0.5 (*n* = 1)	**K372E, I377V, G398A, V405A, Q412L, G429D,** **H438Y, D478A, E513Q, Q557E**	NT	NT	NT	Japan (2010–2013)
Metcalf, 2017 [36]	0.25 (*n* = 6)	I377V, **G406D,** **Q557E**	NT	NR	NR	USA (2015)
Sigaúque, 2018 [48]	0.12 (*n* = 7)	**G398A**	ND	ND	ND	Mozambique (2001–2015)
Yi, 2019 [49]	0.5 (*n* = 2)	**G398A, V405A, Q557E**	NT	NT	NT	South Korea (NI)
Van der Linden, 2020 [50]	0.5 (*n* = 1) 1.0 (*n* = 1)	I377V, F395L, **V405A**, H438Y, V510I, **Q557E**	ND	**V80A**	**A521V**, del719–722, **N723S**, **V726A,** **T526I**	Germany (NI)
Li, 2020 * [51]	2.0 (*n* = 1)	I377V, T720S	E63K	V80A, S147A, S160A	T701P V706A	Hong Kong (2014–2017)
McGee, 2021 [52]	0.25 (*n* = 6)	I377V, **G406D** **G398A,** G627V V510I, I510V **Q557E, L534S** G627V	NT	NT	NT	USA (2015–2017)
Nishiyama, 2022 [53]	0.25 (*n* = 5) 0.5 (*n* = 3)	**G398A, V405A, Q557E**, N575D, S295G, **Q557E**	NT	ND	G719A, R629H	Japan (2017–2018)
Koide, 2022 [54]	0.25 (*n* = 7) 1.0 (*n* = 1)	**V405A, G329V, G398A, G429D,** I377V, **F395L, R433H,** H438Y, V510I, **G648A,** **V405A, Q412L, Q557E**	Defective	**T567I**	T587I, F524V, G719N	Japan (2008–2016)
Chu, 2007 [41]	0.25 (*n* = 1)	ND	ND	ND	ND	Hong Kong (2005)
Ikebe, 2023 [55]	0.25 (*n* = 6) 0.5 (*n* = 1)	I377V, G398A, Q412L, H438Y, **V405A, Q557E,** F395L, R433H, T473M, V510I, G329V, K372E, G429D	ND	T567I	T537P, F541V, N635-, G636-, A621V, A547V	Japan (2014–2021)
Ntozini, 2025 [56]	0.25 (*n* = 1)	New types that are not yet assigned a number	ND	ND	ND	South Africa (2019–2020)
McGuire, 2025 [57]	1.0 (*n* = 1)	**I342V, V475I** **P160S, Y331H** **P140L, N540D**	**A27T, N741A** **V744A, V541I**	**V80A**	**K63E**	England (2016)

ND, not detected; NR, not reported; NT, not tested; NI, not informed. Substitutions strongly associated with penicillin-resistant *S. agalactiae* are shown in boldface. * Described mutation A95D in the *pbp1b* gene.

## 4. *S. agalactiae* with Reduced β-Lactam Susceptibility

The WHO Bacterial Priority Pathogens List (PBPL, 2024) has been fundamental in guiding global policies, research and development, and investments to address the most urgent threats of antibiotic-resistant pathogens, being essential for public health in the prevention and control of antimicrobial resistance [58,59]. *S. agalactiae* is listed in the PBPL as a microorganism with a concerning increase in resistance to β-lactams. *S. agalactiae* is a major cause of pregnancy complications and neonatal morbidity and mortality worldwide, as well as invasive infections in elderly and non-pregnant adults with comorbidities [60]. However, current disease burden data from low- and middle-income countries are sparse, demonstrating that there is still little information on RPS *S. agalactiae* in these countries. The absence of official protocols from federal governments, insufficient funding, and difficulty of access to health facilities for pregnant women, lack of infrastructure, as well as barriers to data sharing may be responsible for the disparities in research publications in low- and middle-income countries.

Several publications have described *S. agalactiae* strains with PRS or resistance to β-lactams in Asia, North America, Africa, and Europe. However, there are no published reports of RPS or penicillin-resistant *S. agalactiae* strains in China—except in Hong Kong [41,51], India, and Central and South America. Detailed characterization of PRS *S. agalactiae* with reduced susceptibility to β-lactams related to mutations in the *pbp2x* was first reported in 2008 [3]. The same research group identified strains with other mutations in *pbp2x*, *pbp2a*, *pbp2b*, and *pbp1a* in the years 2008, 2012, and 2014 [35,44,45]. Also in Japan, other researchers have associated novel mutations in *pbp2x* and *pbp1a* with the reduced susceptibility of *S. agalactiae* strains to penicillin [46,47,53,54,55] (Table 1). These data obtained in Japan demonstrated a significant increase in RPS *S. agalactiae* strains, as well as the diversity of mutations described in various PBPs. It is necessary to monitor penicillin-resistant/reduced susceptibility strains in order to identify new mutations and multidrug-resistant (MDR) strains that could spread globally.

In Canada, different mutations from those described in Japan were identified in *pbp2x*, *pbp2a*, *pbp2b*, and *pbp1a* of RPS *S. agalactiae* strains [42,43]. A total of 28,269 isolates of invasive *S. agalactiae* infections, collected over 20 years in the United States and identified with reduced sensitivity to β-lactams, were detected across all Active Bacterial Core surveillance sites. The results revealed the emergence of first-step *pbp2x* mutations conferring reduced susceptibility to β-lactams among *S. agalactiae* causing invasive disease in the United States [61]. In Mozambique [48], South Korea [49], and Northwest Ethiopia [62], mutations were also identified exclusively in *pbp2x*. In contrast, in England, mutations were identified in all four PBPs-encoding genes, while in Germany they were found in *pbp2x*, *pbp2b*, and *pbp1a* among RPS *S. agalactiae* strains (Table 1).

In Hong Kong, China, the first report of RPS *S. agalactiae* occurred in 2007 [41], and in 2020, a second strain of RPS *S. agalactiae* was isolated [51], exhibiting mutations in five PBPs-encoding genes produced by *S. agalactiae* (*pbp1a*, *pbp1b*, *pbp2a*, *pbp2b*, *pbp2x*; Table 1). In South Korea, a 24-year study assessed trends in penicillin non-susceptibility in *S. agalactiae* and demonstrated a statistically significant increase in both the isolation rate of *S. agalactiae* and penicillin non-susceptibility over time [63]. Data from South Africa in the period 2019–2020 identified one PRS *S. agalactiae* strain. However, the authors were unable to detect the mutation in the *pbp2x* gene due to insufficient sequence quality at the PBP2x locus [56]. Continuous exposure to β-lactam antibiotics can contribute to the accumulation of additional mutations, leading to higher levels of resistance. Therefore, elderly individuals and people with comorbidities, who are at higher risk of invasive disease caused by *S. agalactiae*, should be monitored for increased MIC values to prevent invasive disease caused by PRS *S. agalactiae*. Furthermore, it is necessary to monitor prior antibiotic exposures to avoid increasing the proportion of RPS *S. agalactiae* strains and prevent dissemination in hospital and community settings. Surveillance should be continuous, as strains with reduced sensitivity to β-lactams exhibit a high level of successful adaptation to the selective pressure of these antibiotics.

In addition to the *S. agalactiae* strains identified with reduced susceptibility to β-lactams in different countries, a significant percentage of these bacterial strains also showed resistance to other classes of antimicrobials, being classified as MDR. A pattern of MDR in RPS *S. agalactiae* isolated from pregnant women receiving prenatal care in Ethiopia in 2021 showed a high frequency of 26.7% for resistance to 3 classes of antimicrobials (penicillin, erythromycin, vancomycin) and 6.7% for resistance to 4 classes of antimicrobials (penicillin, erythromycin, vancomycin, and ceftriaxone) [62]. Results in Japan also showed a significant association between RPS and non-susceptibility to fluoroquinolones and macrolide resistance in clinical isolates of *S. agalactiae* [37]. MDR *S. agalactiae* strains were also identified for resistance to β-lactams, tetracycline, fluoroquinolone, vancomycin, and clindamycin in the United States [52]; for penicillin, levofloxacin, erythromycin, clindamycin, and tetracycline in Korea [64]; and for penicillin, erythromycin, clindamycin, and tetracycline [41], in addition to fluoroquinolones and aminoglycosides, in Hong Kong [51]. These data demonstrate a limited repertoire of medications for the effective treatment of RPS *S. agalactiae* infections, highlighting the need for careful drug selection for medical treatment in clinical practice. Therefore, epidemiological research on RPS *S. agalactiae* strains is urgently needed to improve the clinical management of invasive infections caused by this pathogen. Furthermore, invasive RPS *S. agalactiae* isolates carry important virulence and resistance genes, demonstrating the need for population-based genomic surveillance to deepen the understanding of the clinical relevance of these invasive *S. agalactiae* isolates. MDR PRS *S. agalactiae* isolates retained a core virulence gene repertoire (*bibA*, *fbsA*, *fbsB*, *fbsC*, *cspA*, *cfb, hylB*, *scpB*, *lmb,* and the *cyl* operon), supporting an invasive ability similar to that of the other invasive *S. agalactiae* penicillin-susceptible [54].

The main publications on *S. agalactiae* strains with reduced susceptibility to β-lactams performed capsular typing and multilocus sequence typing (MLST) assays to identify the circulating type sequences (STs) in the region. Epidemiological and molecular analyses showed a high diversity of genetic lineages among RPS *S. agalactiae* isolates. Thus, the map of the main capsular types and STs described to date is represented in Figure 2.

In Japan, there was a predominance of RPS isolates belonging to capsular types Ia, Ib, II, III, VI, and VII, classified within ST1, ST3, ST10, ST358, ST458, ST464, and ST1404 [27,28,29]. In Hong Kong, types Ib/ST651 and VI [41,51]; Mozambique, type III/ST109 [48]; Sweden, types III and V [65]; England, type Ia/ST144 [57]; Germany, type Ia/ST23 [50]; Canada, type II/ST2 [42]; and the United States, types Ia/ST23, II/ST22, III/ST19, III/ST109, III/ST828, and V/ST1 [36,52]. Collectively, we found that RPS *S. agalactiae* isolates were frequently detected in adults, mainly belonging to serotype III described in North America, Europe, Asia, and Africa, followed by serotype Ia identified in North America, Europe, and Asia. A genomic study of invasive RPS isolates in Japan confirmed the close genomic relationship of RPS isolates of serotype Ia/ST1, as well as the persistence of these isolates at the site of infection for more than 3 weeks, increasing the risk of sepsis in the elderly [34]. The emergence of evolutionary lineages of invasive RPS isolates of serotypes Ia/ST1 and III/ST1 has also been investigated. Some invasive RPS isolates showed the acquisition of mobile genetic elements associated with the antibiotic resistance gene *mefA-msrD* or *aac*(*6*′)*-aph*(*2*′′), confirming the increasing trend in the occurrence of RPS among *S. agalactiae* isolates from various clinical sources [54].

Although there are few studies worldwide that cite the isolation sites and clinical manifestations of PRS *S. agalactiae* strains, we can verify that the most frequent isolation sites were the respiratory tract [37,44,46], followed by recto-vaginal [62,66], sputum [3,49,53], blood [37,43,47,48,52,54], pus [42,44,47], abscess [50], joint [47], urine [43], ulcer [34], subdural hematoma [35], and wound [51] (Figure 3A). Regarding the clinical manifestations reported in the publications, we observed bacteremia [47,54], pneumonia [44,47,49,53], aneurysm, meningitis, hydrocephalus [34,44,48], invasive infections [36], cancer [44,53], osteomyelitis [43], arthritis, cellulitis and spondylitis [47], diabetes mellitus [53], streptococcal toxic shock syndrome [55], and fistula [52] as the most frequent (Figure 3B).

Current guidelines recommend IAP for pregnant women at risk of transmitting *S. agalactiae* to their newborns, as well as for the treatment of invasive infections in the elderly and adults with comorbidities. The rise in RPS *S. agalactiae* strains will have a direct impact on clinical medicine, leading doctors to adopt alternative antibiotics, such as clindamycin or erythromycin, which have also shown high resistance rates in *S. agalactiae*, as well as increased resistance to fluoroquinolones. Furthermore, hospitals and laboratories may need to improve monitoring of *S. agalactiae* resistance patterns, making changes to treatment protocols based on local resistance trends. Health surveillance programs should track the prevalence of reduced penicillin sensitivity in *S. agalactiae*, with implications for vaccine development and infection control strategies. Increased awareness among healthcare professionals regarding the emergence of resistant strains is crucial for timely diagnosis and treatment, ensuring patient safety, particularly in high-risk populations and in low- and middle-income countries.

However, important questions remain unanswered. Why have countries with large geographic areas, such as China and India, considered important world economies, not yet identified RPS *S. agalactiae* strains, while other countries, also with large geographic areas and important world economies, such as the United States and Canada, have already identified RPS *S. agalactiae*? Similarly, Brazil, as an emerging country, has not yet identified RPS *S. agalactiae*. The core of this issue may be the screening for *S. agalactiae* through official protocols implemented in the countries. The United States has implemented the guidelines of the Centers for Disease Control and Prevention (CDC) and the American College of Obstetricians and Gynecologists (ACOG) for the identification of *S. agalactiae* in pregnant women, while in China, there are no standardized guidelines on *S. agalactiae* screening and prevention [67]. Similarly, Brazil does not have any official protocol for screening pregnant women for *S. agalactiae* through the public health system [68]. Although the Indian national guidelines for Infection Prevention and Control in Healthcare Facilities, adapted from the WHO, recommend a risk-based approach for *S. agalactiae* infection prophylaxis, routine antibiotic administration during labor is not recommended [69]. Furthermore, other problems can greatly contribute to the lack of information on RPS *S. agalactiae* in different countries. Insufficient funding and a lack of research culture mean that many low- and middle-income countries do not prioritize research in their national budgets, and the lack of infrastructure, limited access to equipment, and lack of data archives and computational resources hinder data collection and management, which together are responsible for disparities in information, particularly in low- and middle-income countries.

In conclusion, these findings indicate that the isolation rate of PRS *S. agalactiae* has been rapidly increasing, particularly in Japan. Furthermore, isolation rates of multidrug-resistant RPS *S. agalactiae* were found to be quite high among all clinical isolates of S. agalactiae. Given the emergence of multidrug-resistant *S. agalactiae* strains, susceptibility testing and interdisciplinary collaboration between microbiologists and clinicians are crucial for guiding effective antimicrobial therapy and preventing *S. agalactiae* infections in neonates and adults.

## 5. Future Perspectives

The increasing isolation of PRS *S. agalactiae* strains is concerning. Thus, the effectiveness of β-lactams may be compromised, making it necessary to inquire about the validity of PBPs in future therapies against this pathogen. β-lactams mimic the substrates of PBPs; therefore, understanding how the natural substrates of PBPs or partner proteins maintain reactivity at the catalytic site, even in PBPs from resistant bacteria, may provide relevant information for the development of new compounds [11]. Additionally, new molecules may act as adjuvants to restore or maintain the reactivity of PBPs to traditional β-lactams. The next major challenges will involve describing the mechanisms of how PBPs are activated and regulated, as well as identifying protein partners that will help clarify important questions.

Several substances have been analyzed as non-β-lactam PBP inhibitors. Peptidomimetic boronic acid inhibitors have been studied using *S. pneumoniae* PBP1B as a model enzyme, obtaining promising results. The antibacterial activity of boronates has also shown good results against methicillin-resistant *Staphylococcus aureus* (MRSA; IC50 = 6.9 μM) [70]. Moreover, xeruborbactam, a cyclic boronic acid β-lactamase inhibitor, was able to bind to PBP1A/PBP1B, PBP2, and PBP3 of *Escherichia coli* and *Klebsiella pneumoniae* with IC50s in the range of 40 to 70 μM [71]. Additionally, fragment-based computational strategies have been used to find molecules that non-covalently bind to the catalytic site of PBPs. Molecular docking analyses suggested that quinolones were promising candidates for binding [72]. The creation of a β-peptide oligomer library, through synthetic fermentation, using α-ketoacid-hydroxylamine with a mixture of α-ketoacids and isoxazolidine monomers, led to the identification of peptide 62. Peptide 62 was able to inhibit the growth of *Bacillus subtilis* with a MIC of 5.7 ug/mL and low toxicity (IC50 above 100 ug/mL) [73].

Another strategy is to use inhibitors against other proteins involved in peptidoglycan synthesis. MraY is an integral membrane protein essential for bacterial growth, responsible for catalyzing the first step in the synthesis of membrane-associated peptidoglycan [74]. Different classes of natural inhibitors have targeted MraY with antibacterial activity, such as capuramycin and tunicamycin, in the search for new inhibitors [75]. Future research should also aim to elucidate the functional consequences of the identified *pbp* mutations, including their effects on PBP structure, β-lactam binding affinity, and bacterial fitness. Expanded genomic surveillance across underrepresented regions will be essential to determine whether these mutations arise through independent evolutionary events or clonal dissemination. Moreover, it is important to assess the potential progression from first-step mutations to higher levels of β-lactam resistance. Additional studies should also evaluate the impact of these mutations on the activity of other β-lactams, as well as their possible association with virulence determinants.

## Figures and Tables

**Figure 1 antibiotics-15-00031-f001:**
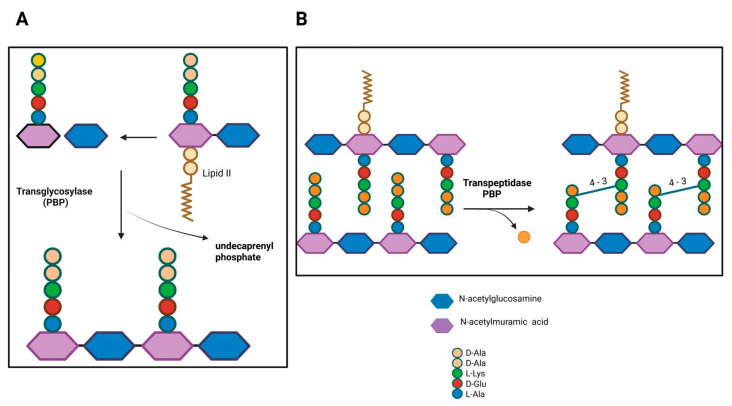
Penicillin-binding proteins in bacterial peptidoglycan synthesis. (**A**) PBPs with transglycosylase activity. (**B**) Transpeptidase PBP activity. D-Ala, D-alanine; L-Lys, L-lysine; D-Glu, D-glutamate; L-Ala, L-alanine. Figure created with BioRender.com.

**Figure 2 antibiotics-15-00031-f002:**
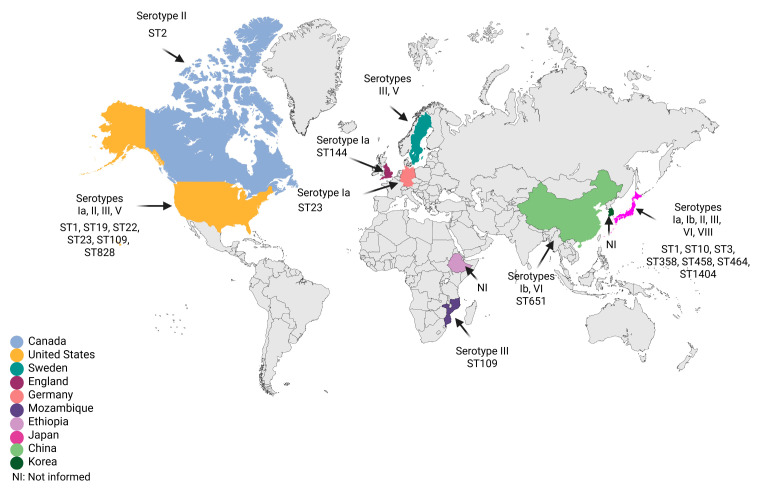
Geographic distribution of reduced susceptibility to penicillin in *Streptococcus agalactiae* strains with available serotypes and sequence types. Figure created with BioRender.com.

**Figure 3 antibiotics-15-00031-f003:**
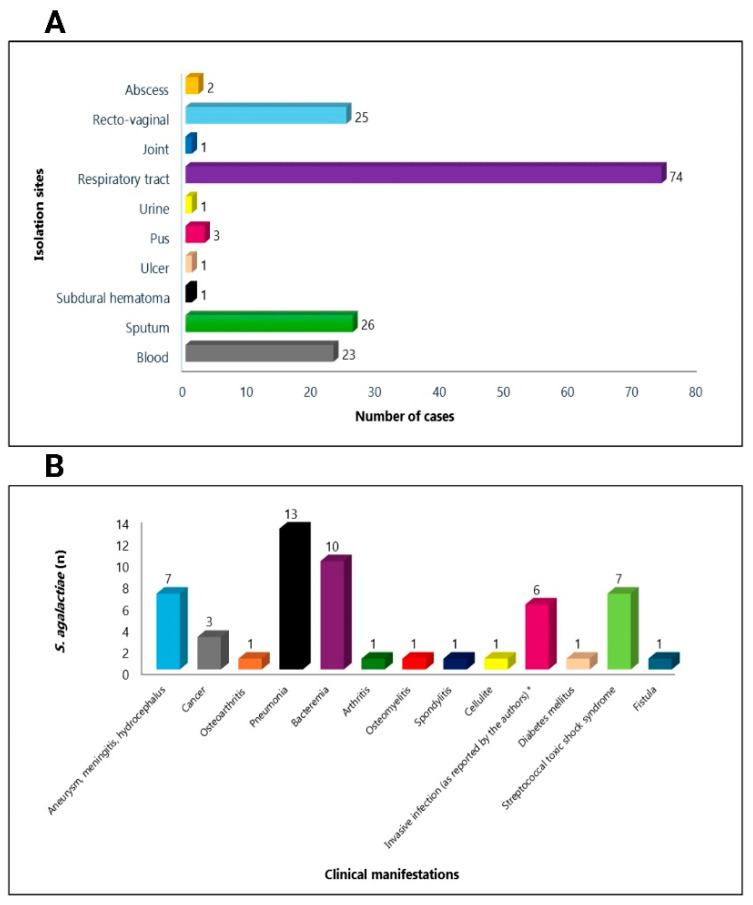
Distribution of the main isolation sites (**A**) and clinical manifestations (**B**) of reduced-penicillin susceptibility of *Streptococcus agalactiae* strains recovered from humans * [36]. Figure created with BioRender.com.

## Data Availability

No new data were created or analyzed in this study. Data sharing is not applicable to this article.

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
