# Peer review of "Penicillin-Binding Proteins in Streptococcus agalactiae and Their Association with Reduced Penicillin Susceptibility: An Overview"

_antibiotics, 2026, doi:10.3390/antibiotics15010031_

Round 1

Reviewer 1 Report

Comments and Suggestions for Authors

The manuscript describes the role of penicillin binding proteins in Streptococcus agalactiae and how penicillin susceptibility is affected by PBP mutations. This review provides an excellent summary on the current state-of-the-art in this topic. Also, it provides all the information required to understand the global problem caused by penicillin-resistant S.agalactiae infections. This manuscript will serve as an outstanding reference in this topic for the future.

Just a few minor comments for the authors:

1) Throughout the manuscript the authors seems to refer to penicillin as a single molecule rather than a group of antibiotics. For example, in line 39 the authors mention "penicillin is..". The authors should specify whether they are referring to penicillin G as "penicillin" or they are intending penicillin as a group, and in this case, use the plular ("penicillins are the first line treatment..").

2) HMM abbreviation should be specified in line 54 rather than 63 as it was done for PBPs in line 53.

3) In the future perspectives section, you mention boronic acids as as potential PBP inhibitors. Although, I completely agree with that, it would be great if the authors provide some data to support this. Are the MIC and IC50 available? Which compounds are the authors referring to? Cyclic boronic acids, peptidomimetic or other boronic acids? 

The english quality is excellent.

Author Response

Thank you for your message and critical comments concerning the manuscript “Penicillin binding protein and Streptococcus agalactiae with reduced penicillin susceptibility” by Ferreira et al. The manuscript submitted to Antibiotics-MDPI (Manuscript ID: antibiotics-4043614) was revised and supplied with new information to improve data quality, as pointed out by the reviewers.

Reviewer #1 comments:

The manuscript describes the role of penicillin binding proteins in Streptococcus agalactiae and how penicillin susceptibility is affected by PBP mutations. This review provides an excellent summary on the current state-of-the-art in this topic. Also, it provides all the information required to understand the global problem caused by penicillin-resistant S.agalactiae infections. This manuscript will serve as an outstanding reference in this topic for the future.

Just a few minor comments for the authors:

1) Throughout the manuscript the authors seems to refer to penicillin as a single molecule rather than a group of antibiotics. For example, in line 39 the authors mention "penicillin is..". The authors should specify whether they are referring to penicillin G as "penicillin" or they are intending penicillin as a group, and in this case, use the plular ("penicillins are the first line treatment..").

Lines 39-40: Penicillins are the first line treatment the first-line antibiotic for the treatment of S. agalactiae infections and is crucial for preventing vertical transmission…

2) HMM abbreviation should be specified in line 54 rather than 63 as it was done for PBPs in line 53.

Lines 70-72: Low molecular weight PBPs (LMM class C) are monofunctional enzymes, such as carboxypeptidases involved in peptidoglycan remodeling, while high molecular weight PBPs (HMM class A and B) are responsible for…

3) In the future perspectives section, you mention boronic acids as as potential PBP inhibitors. Although, I completely agree with that, it would be great if the authors provide some data to support this. Are the MIC and IC50 available? Which compounds are the authors referring to? Cyclic boronic acids, peptidomimetic or other boronic acids?

Lines 389-395: Several substances have been analyzed as non-β-lactam PBP inhibitors. Peptidomimetic boronic acid inhibitors have been studied using S. pneumoniae PBP1B as a model enzyme, obtaining promising results. The antibacterial activity of boronates has also shown good results against methicillin-resistant Staphylococcus aureus (MRSA; IC50 =6.9 μM) [70]. Moreover, xeruborbactam, a cyclic boronic acid β-lactamase inhibitor, was able to bind to PBP1A/PBP1B, PBP2, and PBP3 of Escherichia coli and Klebsiella pneumoniae with IC50s in the range of 40 to 70 μM [71].

Reviewer 2 Report

Comments and Suggestions for Authors

Dear Authors,

I hope you are doing well. After carefully reviewing your manuscript, which provides an extensive overview of penicillin-binding proteins (PBPs) in Streptococcus agalactiae and their association with reduced penicillin susceptibility (RPS), I would like to share several observations that may help strengthen the clarity and scientific impact of the work. The manuscript compiles structural, functional, and genetic information on PBPs, summarizes global reports of isolates with decreased susceptibility, and presents a broad collection of mutations identified across different regions. The topic is highly relevant, given the increasing number of studies reporting elevated MICs in S. agalactiae, and the review has strong potential to become a valuable reference on the molecular basis of emerging β-lactam resistance in this species.

The title is not fully clear or grammatically complete. It should explicitly indicate that the manuscript is a review and better reflect the focus on PBPs and reduced penicillin susceptibility.

The introduction contains repeated general information on the clinical aspects of S. agalactiae. This section should be shortened and more directly connected to the central topic of reduced penicillin susceptibility (RPS). The rationale for the review should also be strengthened by clearly outlining the existing knowledge gap and explaining why a synthesis of PBP-associated resistance mechanisms is needed.

The section on Penicillin-Binding Proteins is overly long and reads more like a general biochemistry chapter. Condensing this part by focusing on the PBPs most relevant to RPS would improve clarity. Additionally, the distinctions among PBP2X, PBP1A, PBP2A, and PBP2B should be more explicitly highlighted, emphasizing their experimentally supported functional roles.

The Figure 1 legend is too long and includes explanations that would be more appropriate in the main text. The legend should be concise and include definitions for all abbreviations.

Regarding mutations in PBPs, several mutation notations appear to contain typographical errors; for example, “1377V” should likely be “I377V,” and “V5101” should be corrected to “V510I.”

Table 1 is very large and difficult to interpret in its current form. The abbreviations ND, NT, and NR should be defined directly beneath the table. Additionally, some mutation names appear inconsistent or contain typographical errors and should be verified. For improved readability, it is recommended to split the table into two smaller tables.

The source of the data used to construct Figures 2 and 3 should be clearly stated. The categories in Figure 3 require clearer definition, and it would be helpful to indicate the number of isolates represented in each region or clinical category.

The discussion would benefit from a deeper comparative analysis across studies. Important limitations are not addressed, including uneven geographic representation, limited data from low-income regions, and the potential for reporting bias. The section could also be strengthened by further elaborating on the clinical implications of increasing RPS in S. agalactiae.

I hope these suggestions contribute to further improving the clarity and impact of your work.

Regards

Author Response

Thank you for your message and critical comments concerning the manuscript “Penicillin binding protein and Streptococcus agalactiae with reduced penicillin susceptibility” by Ferreira et al. The manuscript submitted to Antibiotics-MDPI (Manuscript ID: antibiotics-4043614) was revised and supplied with new information to improve data quality, as pointed out by the reviewers.

1- The title is not fully clear or grammatically complete. It should explicitly indicate that the manuscript is a review and better reflect the focus on PBPs and reduced penicillin susceptibility.

Response: Penicillin-Binding Proteins in Streptococcus agalactiae and their association with reduced penicillin susceptibility: an overview

2- The introduction contains repeated general information on the clinical aspects of S. agalactiae. This section should be shortened and more directly connected to the central topic of reduced penicillin susceptibility (RPS). The rationale for the review should also be strengthened by clearly outlining the existing knowledge gap and explaining why a synthesis of PBP-associated resistance mechanisms is needed.

Response: The introduction has been revised as suggested by the reviewer.

Lines 37-66: Streptococcus agalactiae (group B Streptococcus, GBS) is widely recognized as a leading cause of neonatal sepsis, and invasive disease in elderly or people with comorbidities [1]. Penicillins are the first line treatment the first-line antibiotic for the treatment of S. agalactiae infections and is crucial for preventing vertical transmission to the newborn through intrapartum antibiotic prophylaxis (IAP) [2]. However, isolation of S. agalactiae presenting reduced penicillin susceptibility (RPS) have been reported since 2008 [3] and become a matter of concern worldwide.

The emergence and progression of antimicrobial resistance constitute one of the most significant dangers to public health, as highlighted in clinical investigations supervised by the World Health Organization (WHO). RPS S. agalactiae strains are defined based on antimicrobial susceptibility testing. Both the European Committee on Antimicrobial Susceptibility Testing (EUCAST) and the Clinical and Laboratory Standards Institute (CLSI) provide standardized guidelines for performing antimicrobial susceptibility testing and for interpreting clinical breakpoints for S. agalactiae. EUCAST has established an epidemiological cut-off value (ECOFF) for penicillin in S. agalactiae of 0.125 mg/L, as well as zone diameter breakpoints of > 18 mm [4]. According to CLSI, isolates of S. agalactiae with RPS are defined as those with a minimum inhibitory concentration (MIC) > 0.12 µg/mL [5]. Isolates with MICs above this threshold are considered to have acquired resistance mechanisms.

In S. agalactiae, the RPS has been attributed to the acquisition of mutations in genes encoding the penicillin-binding proteins (PBPs) [6,7], enzymes that catalyze the biosynthesis of bacterial cell wall peptidoglycan. These mutations result in the production of PBPs with reduced affinity for β-lactam binding. However, significant gaps remain in our understanding of the functional impact of individual or combined PBP mutations, therapeutic failure resulting from small increases in MIC, and the evolutionary dynamics that may drive the worldwide spread of RPS S. agalactiae strains.

Therefore, this review aimed to gather available epidemiological data and current knowledge on the most relevant PBP mutations associated with RPS and penicillin resistance (PR) in S. agalactiae strains, contributing to identifying priorities for future research and improving the clinical management of S. agalactiae infections.

3- The section on Penicillin-Binding Proteins is overly long and reads more like a general biochemistry chapter. Condensing this part by focusing on the PBPs most relevant to RPS would improve clarity. Additionally, the distinctions among PBP2X, PBP1A, PBP2A, and PBP2B should be more explicitly highlighted, emphasizing their experimentally supported functional roles.

Lines 70-176: PBPs comprise two classes based on their molecular weight and enzymatic activity (glycosyltransferase and/or transpeptidase). Low molecular weight PBPs (LMM class C) are monofunctional enzymes, such as carboxypeptidases involved in peptidoglycan remodeling, while high molecular weight PBPs (HMM class A and B) are responsible for peptidoglycan polymerization and its insertion into the pre-existing cell wall [8]. The N-terminal domain of HMM PBPs class A is responsible for glycosyltransferase activity, catalyzing the elongation of non-crosslinked glycan chains. After lipid II (disaccharide-pentapeptide attached to the pyrophosphate-tethered undecaprenyl tail) is inverted to the periplasmic side, glycosyltransferases polymerize the sugar chains (Figure 1A); while the penicillin-binding C-terminal domain has transpeptidase activity, catalyzing the cross-linking of peptides between two adjacent glycan chains. PBP recognizes the terminal portion of D-alanine, catalyzing the attack of the carbonyl group of the penultimate D-alanine by the lateral amino group at position (3) of an adjacent chain (4–3 cross-link; Figure 1B). In class B, the N-terminal domain plays a role in cell morphogenesis through interaction with other proteins involved in the cell cycle [9].

PBPs possess a highly stable covalent complex through the serine in their active site, where penicillin and other β-lactams bind irreversibly, forming a stable acyl-enzyme intermediate that permanently inactivates the enzyme [10,11]. The catalytic serine attacks the carbonyl group of the β-lactam ring, promoting ring opening and the formation of a stable covalent acyl-enzyme complex that impairs transpeptidation, leading to peptidoglycan cross-linking failure and bacterial death [12]. Altered PBPs, resulting from multiple homologous recombination events between genes of closely related species (mosaic genes), combined with additional point mutations, have been described in pneumococci and oral streptococci, where amino acid substitutions represent more than 10% of the primary sequence [7]. Identifying the amino acid alterations involved in resistance, combined with structural information, will provide a better understanding of the enzymatic function of PBPs and the development of new inhibitors. However, mosaic sequences of pbp genes represent a challenge for classification and organization. Comparison of nucleotide sequences from susceptible strains reveals that these sequences exhibit levels of polymorphism similar to other loci, with one or two amino acid substitutions along the protein [6]. Mosaic pbp genes also exhibit sequence blocks that approximate the non-mosaic alleles in PBP2B, PBP1A, and PBP2X [6,13,14]. These blocks can encompass various regions of the transpeptidase domain or even a large part of the extracellular domain, the magnitude of which suggests that these divergent sequence blocks originate from other streptococcal species to escape protein selective pressure [6,15,16]. S. agalactiae produces three bifunctional class A PBPs (PBP1A, PBP1B and PBP2A) and two monofunctional class B PBPs (PBP2X and PBP2B) [17]. However, few studies report the biological functions and differences between PBPs in S. agalactiae.  Therefore, to fill this gap, the general differences and biological functions of the PBPs analyzed in different microorganisms will be described.

The origin of sequence blocks in pbp mosaic genes is still unknown, with some exceptions for PBP2X. Study revealed that a family of mosaic pbp2x genes occurred in Streptococcus pneumoniae, Streptococcus mitis and Streptococcus oralis demonstrating independent inter- and intraspecies recombination events [18]. These data gave rise to the hypothesis that commensal streptococci, such as S. mitis and S. oralis acquired resistance through point mutations by exposure to β-lactams treatment for diseases, being subsequently exchanged between related streptococcal species. PBP2X depletion in S. pneumoniae promoted elongated and swollen cells with pointed ends [19]. Resolution 3D structured illumination microscopy (3D-SIM) showed the exceptional localization of PBP2X in the centers of septa in cells at intermediate to late stages of division, suggesting that the proteoglycan transpeptidase cross-linking activity of PBP2X has a biological function during septum formation and proteoglycan remodeling [20]. Phylogenetic analysis showed that Streptococcus pyogenes strains of the same type emm and the same amino acid substitution in PBP2X were clonally related by descent, indicating that strains with alterations in PBP2X can spread to new hosts and cause invasive infections [21].

PBP1A, also called PonA, and PBP1B proteins contain a non-catalytic domain involved in the regulation of glycosyltransferase and transpeptidase activities, and are essential for cell viability in S. agalactiae and S. pneumoniae [10,17,22]. PBP1A is preferentially involved in proteoglycan synthesis during cell elongation and plays an important role in the resistance of S. agalactiae to antimicrobial peptides, potentially representing a widespread mechanism for evading the human innate immune response [23]. PBP1A was important for lung proliferation and resistance to killing by polymorphonuclear leukocytes and alveolar macrophages in a neonatal mouse model of S. agalactiae infection [24]. The mutant S. agalactiae strain (∆ponA) also showed reduced virulence in a murine sepsis model compared to the wild-type strain [25]. PBP1B is involved in septation, being found in high concentrations in the mid-cell region during cell constriction. FtsN, a divisome-specific protein, binds to PBP1B, stimulating its glycosyltransferase biosynthetic activity, contributing to the synthesis and overall regulation of septal peptidoglycan during cell division [26].

PBP2 is involved in cell division, particularly at its location in the septum. PBP2A is acquired by certain bacteria, such as methicillin-resistant Staphylococcus aureus (MRSA), where the first crystal structure of a soluble form of PBP2A was published in 2002 [27]. Crystallographic analyses showed that PBP2A, mecA gene product, can maintain transpeptidase activity and resist β-lactam binding, maintaining a closed conformation of the active site. Thus, interaction with a second molecule (e.g., muramic acid, peptidoglycan, or ceftaroline) at an allosteric site induced a conformational change, allowing the active site to open for catalysis [28]. The ability of ceftaroline, a β-lactam antibiotic, to stimulate the allosteric opening of the active site, leading to the inactivation of PBP2A by a second β-lactam, will allow future studies to obtain allosteric inhibitors based on the structure of PBP2A.

PBP2B is a class B specific to cell division with transpeptidase activity [29]. In S. agalactiae, researchers have identified that the polymerization of the glycan chain by the glycosyltransferase activity of RodA, a well-characterized SEDS (shape, elongate, and divide) family protein, is activated by PBP2B generating the production of longer peptidoglycan fragments and faster depletion of the substrate lipid II [30].

 Interestingly, Nishimoto and co-authors suggested that resistance to β-lactams in pathogenic streptococci is restricted to naturally competent species through intra- and interspecific recombination due to in vivo fitness trade-offs from de novo mutations in PBPs. The authors demonstrated that serially transformed recombinant strains efficiently integrate into the DNA of resistant oral streptococci, acquire resistance and tolerance to -lactams, and maintain virulence in an in vivo model, indicating that de novo mutations are of primary relevance [31]. However, other components of the peptidoglycan synthesis machinery likely interact with PBPs directly or indirectly and may be modified in response to changes in the PBPs. All these facts contribute to the difficulties in experimentally verifying the impact of individual mutations within PBPs on penicillin resistance. The availability of larger amounts of genomic data for RPS and commensal streptococci will facilitate obtaining more information about the complex genetic network between species and will unravel mutations in components involved in the expression of penicillin resistance. These observations raise questions about the different roles that PBPs may play in cell wall formation and the possibility that their functions may vary slightly depending on the nature of the bacterium, the local environment of the cell, and the stage of the cell cycle.

4- The Figure 1 legend is too long and includes explanations that would be more appropriate in the main text. The legend should be concise and include definitions for all abbreviations.

Response: The Figure 1 legend has been corrected.

5- Regarding mutations in PBPs, several mutation notations appear to contain typographical errors; for example, “1377V” should likely be “I377V,” and “V5101” should be corrected to “V510I.”

Response: Table 1 has been revised.

6- Table 1 is very large and difficult to interpret in its current form. The abbreviations ND, NT, and NR should be defined directly beneath the table. Additionally, some mutation names appear inconsistent or contain typographical errors and should be verified. For improved readability, it is recommended to split the table into two smaller tables.

Response: Table 1 became misshapen when transferred to the Antibiotics Journal Template. We adjusted the table 1 so that it is not too long. The abbreviations ND, NT, and NR were defined directly below the table, and the mutation names were checked for typographical errors.

7-The source of the data used to construct Figures 2 and 3 should be clearly stated. The categories in Figure 3 require clearer definition, and it would be helpful to indicate the number of isolates represented in each region or clinical category.

Response: The source of the data used to construct Figures 2 and 3 was indicated in the text. The categories in Figure 3 were revised with a clearer definition, indicating the number of isolates represented in each clinical category.

8- The discussion would benefit from a deeper comparative analysis across studies. Important limitations are not addressed, including uneven geographic representation, limited data from low-income regions, and the potential for reporting bias. The section could also be strengthened by further elaborating on the clinical implications of increasing RPS in S. agalactiae.

Lines 227-232: However, current disease burden data from low- and middle-income countries are sparse, demonstrating that there is still little information on RPS S. agalactiae in these countries. The absence of official protocols from federal governments, insufficient funding, and difficulty of access to health facilities for pregnant women, lack of infrastructure, as well as barriers to data sharing the data may be largely responsible for the disparities in research publications in low- and middle-income countries.

Lines 348-369: However, important questions remain unanswered. Why countries with large geographic areas, China and India, considered important world economies, have not yet identified RPS S. agalactiae strains, while other countries, also with large geographic areas and important world economies, United States and Canada, have already identified RPS S. agalactiae? Similarly, Brazil, as an emerging country, has not yet identified RPS S. agalactiae. The core of this issue may be the screening for S. agalactiae through official protocols implemented in the countries. The United States has implemented the guidelines of the Centers for Disease Control and Prevention (CDC) and the American College of Obstetricians and Gynecologists (SCOG) for the identification of S. agalactiae in pregnant women, while China there are no standardized guidelines on S. agalactiae screening and prevention in China [67]. Similarly, Brazil does not have any official protocol for screening pregnant women for S. agalactiae through the public health system [68]. Although the Indian national guidelines for Infection Prevention and Control in Healthcare Facilities, adapted from the WHO, recommend a risk-based approach for S. agalactiae infection prophylaxis, routine antibiotic administration during labor is not recommended [69]. Furthermore, other problems can greatly contribute to the lack of information on RPS S. agalactiae in different countries. Insufficient funding and a lack of research culture mean that many low- and middle-income countries do not prioritize research in their national budgets; and the lack of infrastructure, limited access to equipment, lack of data archives and computational resources that hinder data collection and management, which together are responsible for disparities in research publications in low- and middle-income countries.

Reviewer 3 Report

Comments and Suggestions for Authors

Thanks for submitting the manuscript of reviewing an interesting topic on Streptococcus agalactiae reduced penicillin susceptibility, especially emphasize the PBP mutations, global epidemiology, and clinical implications. In general, this manuscript is well organized. However, it can be improved in some aspects. 

Major comments:

  1. It will be helpful to present a deeper mechanistic explanation of how specific PBP2X mutations alter beta-lactam binding.  Cite some papers from molecular modeling or comparative structure studies.
  2. The epidemiology section needs to be extended. You discussed the findings from different countries, but lack of data from China and India, which are two large countries could provide more useful examples. Review more examples from these two countries will strengthen you study. 

Minor comments:

  1. Ref 10 and Ref 15 are duplicates.
  2. Line 57-62 should be in the Introduction section
  3. Line 175, "encoding" not "enconding"

Author Response

Thank you for your message and critical comments concerning the manuscript “Penicillin binding protein and Streptococcus agalactiae with reduced penicillin susceptibility” by Ferreira et al. The manuscript submitted to Antibiotics-MDPI (Manuscript ID: antibiotics-4043614) was revised and supplied with new information to improve data quality, as pointed out by the reviewers.

  1. It will be helpful to present a deeper mechanistic explanation of how specific PBP2X mutations alter beta-lactam binding. Cite some papers from molecular modeling or comparative structure studies.

Lines 201-208: Important amino acid substitutions in the active site of PBP2x have been established by kinetic and structural studies. Mutation in two residues in the serine region downstream of the active site altered the orientation of the serine hydroxyl group, decreasing the affinity with β-lactams [38] Furthermore, high-resolution analysis of PBP2X and multiple sequence alignments demonstrated an amino acid cluster lining a small water-containing cavity, presumably involved in PBP2X resistance to β-lactam antibiotics. Site-directed mutation in these amino acids significantly reduced the affinity for β-lactams [39].

  1. The epidemiology section needs to be extended. You discussed the findings from different countries, but lack of data from China and India, which are two large countries could provide more useful examples. Review more examples from these two countries will strengthen you study.

Response: To date, no publications from China and India have been found with S. agalactiae strains resistant or with reduced susceptibility to penicillin. Epidemiological data from China and India are from penicillin-sensitive S. agalactiae strains, with the exception of two publications from Hong Kong (Chu et al., 2007; Li et al., 2020), which are cited in Table 1 and in the text.

Lines 234-236: …However, there are no published reports of RPS or penicillin-resistant S. agalactiae strains in China - except in Hong Kong [41,51], India, Central and South America.

Lines 348-369: However, important questions remain unanswered. Why countries with large geographic areas, China and India, considered important world economies, have not yet identified RPS S. agalactiae strains, while other countries, also with large geographic areas and important world economies, United States and Canada, have already identified RPS S. agalactiae? Similarly, Brazil, as an emerging country, has not yet identified RPS S. agalactiae. The core of this issue may be the screening for S. agalactiae through official protocols implemented in the countries. The United States has implemented the guidelines of the Centers for Disease Control and Prevention (CDC) and the American College of Obstetricians and Gynecologists (SCOG) for the identification of S. agalactiae in pregnant women, while China there are no standardized guidelines on S. agalactiae screening and prevention in China [67]. Similarly, Brazil does not have any official protocol for screening pregnant women for S. agalactiae through the public health system [68]. Although the Indian national guidelines for Infection Prevention and Control in Healthcare Facilities, adapted from the WHO, recommend a risk-based approach for S. agalactiae infection prophylaxis, routine antibiotic administration during labor is not recommended [69]. Furthermore, other problems can greatly contribute to the lack of information on RPS S. agalactiae in different countries. Insufficient funding and a lack of research culture mean that many low- and middle-income countries do not prioritize research in their national budgets; and the lack of infrastructure, limited access to equipment, lack of data archives and computational resources that hinder data collection and management, which together are responsible for disparities in research publications in low- and middle-income countries.

  1. Ref 10 and Ref 15 are duplicates.

Response: The duplicated reference was removed, and the reference list has been updated accordingly.

  1. Line 57-62 should be in the Introduction section

Response: The sentence has been relocated to the Introduction section as suggested.

  1. Line 175, "encoding" not "enconding"

Line 254: …mutations were identified in all four PBPs-encoding genes, while in Germany they were…

Round 2

Reviewer 2 Report

Comments and Suggestions for Authors

I appreciate your responses to my observations. I wish you success in the publication of your manuscript.